# The structure of plant photosystem I super-complex at 2.8 Å resolution

**Yuval Mazor\*, Anna Borovikova, Nathan Nelson\***

Department of Biochemistry, The George S. Wise Faculty of Life Sciences, Tel Aviv University, Tel Aviv, Israel

**Abstract** Most life forms on Earth are supported by solar energy harnessed by oxygenic photosynthesis. In eukaryotes, photosynthesis is achieved by large membrane-embedded super-complexes, containing reaction centers and connected antennae. Here, we report the structure of the higher plant PSI-LHCI super-complex determined at 2.8 Å resolution. The structure includes 16 subunits and more than 200 prosthetic groups, which are mostly light harvesting pigments. The complete structures of the four LhcA subunits of LHCI include 52 chlorophyll a and 9 chlorophyll b molecules, as well as 10 carotenoids and 4 lipids. The structure of PSI-LHCI includes detailed protein pigments and pigment–pigment interactions, essential for the mechanism of excitation energy transfer and its modulation in one of nature's most efficient photochemical machines.

## Introduction

Oxygenic photosynthesis, in which the conversion of sunlight into chemical energy by plants, green algae, and cyanobacteria, occurs, underpins the survival of virtually all life forms. By producing oxygen and assimilating carbon dioxide into organic matter, this process determines, to a large extent, the composition of our atmosphere and provides essential food and fuel. Light photons are captured by pigments in very large membrane–bound complexes and the excitation energy is utilized to form NADPH and ATP. Two reaction centers, photosystem II (PSII) and photosystem I (PSI), drive this electron transport chain. PSII oxidizes water to produce oxygen and reduce membrane-embedded quinones. The reduced quinones are then utilized by the cytochrome $b_6f$ complex to produce a proton gradient across the membrane and to reduce the small copper protein plastocyanin (PC), the electron donor of PSI. After an additional photon is absorbed by any of the 200 antenna pigments of PSI, its energy migrates through this large network of connected pigments and eventually oxidizes P700, a special chlorophyll pair located at the center of PSI. The electron removed from P700 by this oxidation event migrates along an internal electron transport chain and finally reduces ferredoxin (Fd), the final electron acceptor of PSI. Reduced Fd is utilized by several cellular pathways, mainly the reduction of NADP to NADPH. This NADPH and ATP generated by the thylakoid ATP-synthase complex powers the Calvin cycle to produce carbohydrates.

Oxygenic photosynthesis evolved over 3 billion years ago in cyanobacteria (*Blankenship, 1992*; *Barber, 2004*; *Nelson, 2013*). Later, approximately 1.5 billion years ago, the first photosynthetic eukaryotes appeared, eventually evolving into land plants roughly 0.5 billion years ago. The basic building blocks of photosynthesis are remarkably conserved. The architectures of both photosystems have been determined by numerous techniques, but X-ray crystallography has provided the most detailed structural information for the four large membrane complexes catalyzing oxygenic photosynthesis (*Nelson and Ben-Shem, 2004*; *Nelson and Yocum, 2006*; *Croce and van Amerongen, 2013*; *Nelson and Junge, 2015*). Structures at the highest resolution has been obtained for thermophilic cyanobacteria, but representative structures from eukaryotic chloroplasts, especially plants, are scarce (*Jordan et al., 2001*; *Ben-Shem et al., 2003*; *Kurisu et al., 2003*;

\*For correspondence: yuval. mazor@gmail.com (YM); nelson@ post.tau.ac.il (NN)

Competing interests: The authors declare that no competing interests exist.

**eLife digest** Most plants, green algae and some bacteria use a process called photosynthesis to convert energy from sunlight into the chemical energy they need to survive and grow. With this energy, these organisms use carbon dioxide and water to create organic matter and release oxygen into the atmosphere. Therefore, photosynthesis plays a major role in providing the basis for life on earth.

During photosynthesis, molecules of pigments known as chlorophyll and carotenoid capture the light energy. These pigments are contained within large groups (or 'complexes') of proteins that sit in membrane structures within cells. Two of the protein complexes—called photosystem I and LHCI—interact with each other to form a 'supercomplex' that transfers energy to a small protein called ferredoxin. To achieve this, the light energy captured by pigment molecules is transferred to other pigment molecules so that the energy is funneled towards the center of photosystem I.

Mazor et al. used a technique called X-ray crystallography to create a very detailed three-dimensional model of photosystem I and LHCI from pea plants. The model shows how the twelve proteins of photosystem I are arranged in relation to the four proteins of the LHCI complex. The super-complex contains more than 200 other molecules, which are mostly chlorophylls and carotenoids. Of these, 61 chlorophyll molecules and ten carotenoid molecules are found in LHCI.

The model also provides detailed information about how the pigments interact with each other and with the proteins in the supercomplex. Mazor et al.'s detailed model may help us to understand how these interactions allow photosystem I to harvest light energy with almost 100% efficiency, and aid efforts to develop new technologies that harness light.

*Stroebel et al., 2003*; *Ferreira et al., 2004*; *Loll et al., 2005*; *Amunts et al., 2007*, *2010*; *Umena et al., 2011*). Here, we report the structure of plant PSI super-complex at high resolution.

## Results and discussion

### Structure determination

The crystal structure of plant PSI was first reported at 4.4 Å resolution (*Ben-Shem et al., 2003*) and has been improved up to 3.3 Å resolution in the last decade (PDB 2WSC). This PSI preparation was limited to pea plants from the variety Alaska, and good crystals were hard to come by (*Amunts et al., 2007*, *2010*). Therefore, we screened for new robust crystals that are abundant, stable, and much more uniform. The new crystals could be obtained from several pea plants varieties, a large proportion of them diffracted to 3 Å with several yielding higher resolutions. In contrast to the $P2_1$ symmetry of the previous crystal, the current crystal belonged to higher symmetry space group $P2_12_12_1$. The organization of the PSI unit within the new crystal was also markedly different. In the $P2_1$ crystal the PSI-LHCI complex was organized as parallel layers in which the iron-sulfur clusters $F_X$, $F_A$ and $F_B$ face the adjacent P700 (*Figure 1A*). The complexes inside the new crystal lattice were serially arranged in a crissed-crossed manner in which the polarity of each PSI unit contrasts another (*Figure 1B*). Consequently, the current crystals generated no net voltage (data not shown), whereas a voltage of up to 50 V was recorded (*Toporik et al., 2012*) upon illumination of dried $P2_1$ crystals placed on electron conductive material.

The extreme size and complexity of the PSI assembly was a major obstacle for accurate and bias-free modeling. The best way to eliminate model bias in X-ray crystallography is to utilize experimentally measured phase information. Using the new, highly stable crystal form of PSI we were able to measure the weak native anomalous signal from the iron, sulfur, and phosphate atoms in the complex. Starting with a minimal model containing only the three natively bound iron-sulfur clusters, the entire structure was eventually re-built with more than 35,000 atoms (*Figure 2* and *Figure 2—figure supplement 1*, see 'Material and methods' section for details).

### Core subunits: PC binding site and implications for the state II PSI complex

The structure of plant PSI includes 12 core subunits bound with four light-harvesting proteins comprising the LHCI antenna complex. The entire complex contains 214 prosthetic groups, including

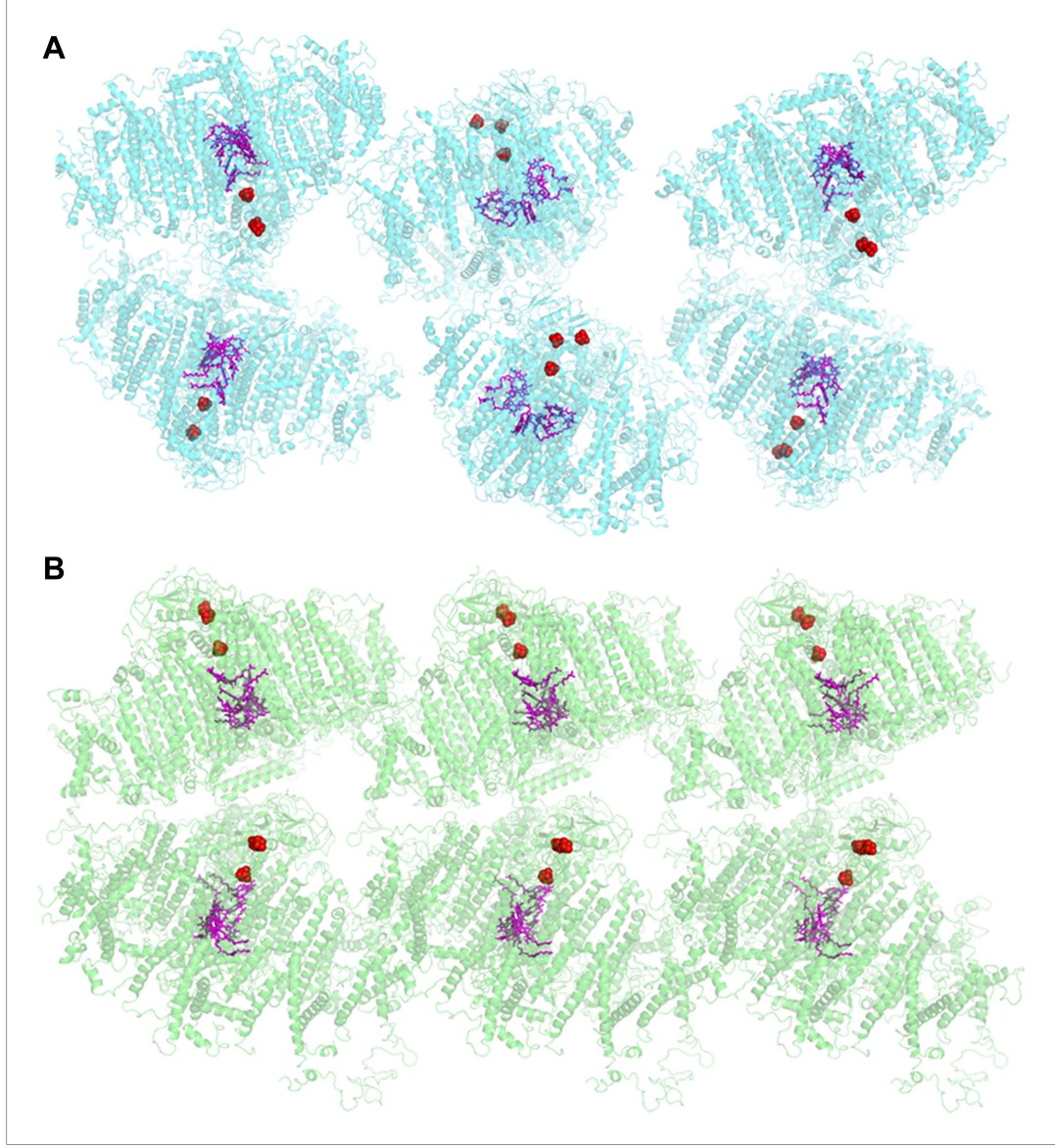

**Figure 1**. Comparison of two PSI-LHCI crystal lattices. (**A**) The previous PSI-LHCI crystal in the P2$_1$ space group with a layered arrangement of the complex. This arrangement is capable of generating extremely high voltages upon illumination. (**B**) The new crystal lattice in the P2$_1$2$_1$2$_1$ space group. Iron sulfur clusters are colored in red and the pigments of the internal electron transport chain in magenta (chlorophylls) and blue (quinones). PSI-LHCI complexes are arranged in a crissed-crossed manner from left to right.

156 chlorophylls (nine assigned as chlorophyll b), 32 carotenes, and 14 lipids, many of them located at key contact points of the complex.

The core photosynthetic reaction centers have remained virtually unchanged over the entire 2 billion years of their evolution (*Jordan et al., 2001*; *Ben-Shem et al., 2003*; *Amunts et al., 2010*). Instead, the evolution of PSI is marked by the loss and gain of whole subunits from the complex (*Scheller et al., 2001*; *Nelson, 2011*; *Nelson and Junge, 2015*). Compared to our previous model (PDB 2WSC), the root-mean-square deviation (rmsd) between the plant and cyanobacterial core (PDB 1JB0) decreased from 1.1 Å to 0.55 Å. The majority of the changes made in the core subunits involved the configuration of extramembrane loops, which now closely resemble the cyanobacterial configuration. The exceptions to this role are found at the anchor points of LHCI to the core (discussed below) and at the interfaces between plant-specific subunits, such as the PsaH–PsaL

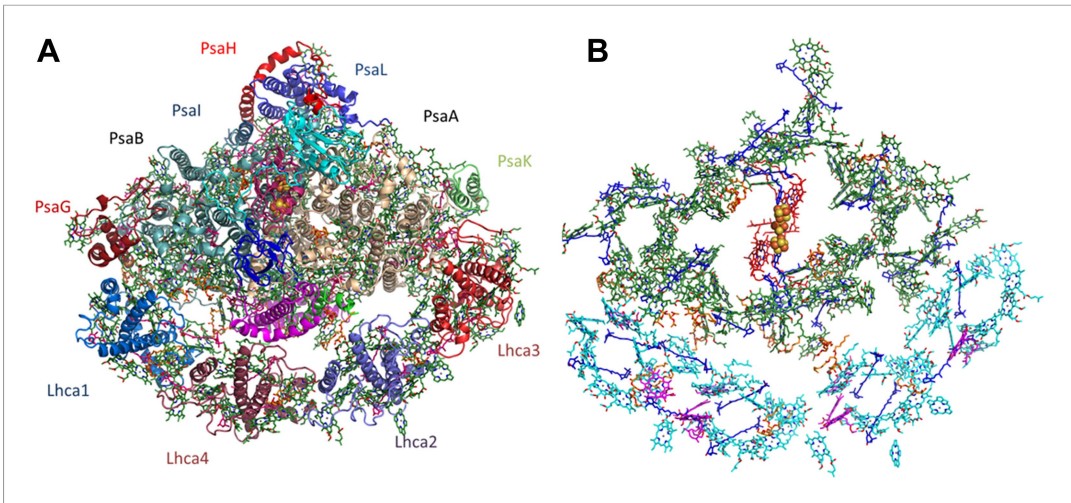

**Figure 2**. Overall structure and organization of the plant PSI-LHCI supercomplex. (**A**) A view from the stromal side of the membrane of PSI-LHCI with the PsaL subunit pointing up. The PsaF and PsaJ subunits connecting in the middle of LHCI are colored in magenta and green, respectively. The three subunits of the stromal ridge, PsaC, PsaD, and PsaE, can be seen in the middle of the complex, colored cyan, pink, and blue, respectively. The two iron-sulfur clusters of PsaC can be distinguished as yellow and orange clusters in the middle of the complex. (**B**) Pigment organization in PSI-LHCI. The central pigments of the internal electron transport chain are colored red, chlorophylls of the core antenna green, chlorophyll a in LHCI in cyan, and chlorophyll b in magenta. Carotenoids, which are distributed throughout the complex, are colored in blue and lipids in key connecting points and conserved positions in the core, in orange.

The following figure supplement is available for figure 2:

**Figure supplement 1**. Phasing substructure and electron density maps.

interaction (*Figure 3A*). The dramatic change from trimer to monomeric organization that occurred in eukaryotes, was triggered by the addition of the PsaH subunit (*Ben-Shem et al., 2003*). A new configuration for PsaH shows that this subunit binds four other core subunits. Starting from the stromal side of the membrane, the N-terminus is tightly tucked between the N-terminus of PsaD and a eukaryotic-specific loop in the PsaL subunit. PsaH then enters the membrane surrounding PsaL to prevent PSI trimerization and associates with PsaI and PsaB via mostly hydrophobic interactions (*Figure 3A*).

Eukaryotes can modulate the distribution of excitation energy transfer between their two photosystems via a mechanism called state transitions (*Lunde et al., 2000*; *Bellafiore et al., 2005*; *Rochaix, 2011*; *Rochaix et al., 2012*). Under state II conditions, PSI associates with a mobile pool of the LHCII antennae, which increases its absorbance cross-section (*Kargul et al., 2005*; *de Bianchi et al., 2010*). Genetic studies suggest that PsaH, PsaL, and PsaK play important roles in this process (*Scheller et al., 2001*; *Zhang and Scheller, 2004*). Electron microscopy studies have identified the binding site of the additional antennae complexes along the PsaL/PsaH-PsaK side (*Kargul et al., 2005*; *Kouril et al., 2005*). A new chlorophyll bound by PsaH was identified at the current resolution. This new chlorophyll, together with pigments bound by PsaL, probably participates in energy transfer into the core (*Figure 3A*), suggesting that PsaH is not simply a 'landing pad' for LHCII, but is also important for energy transfer into the core during state II. Additional pigments bound by PsaA in close proximity to subunit PsaK (the structure of which is now almost completely defined) provided the first accurate description of this binding site and suggest a mechanism for energy transfer into the core antenna through the PsaK side (*Figure 3—figure supplement 1*).

On the luminal side of the membrane, the new position of the N-terminus of PsaH suggests that this subunit also directly contributes to PC binding. One of the distinguishing characteristics of the eukaryotic PSI is the stable complex it forms with its electron donor PC, which results in a thousand-fold acceleration of the electron transfer rate (*Bottin and Mathis, 1985*).

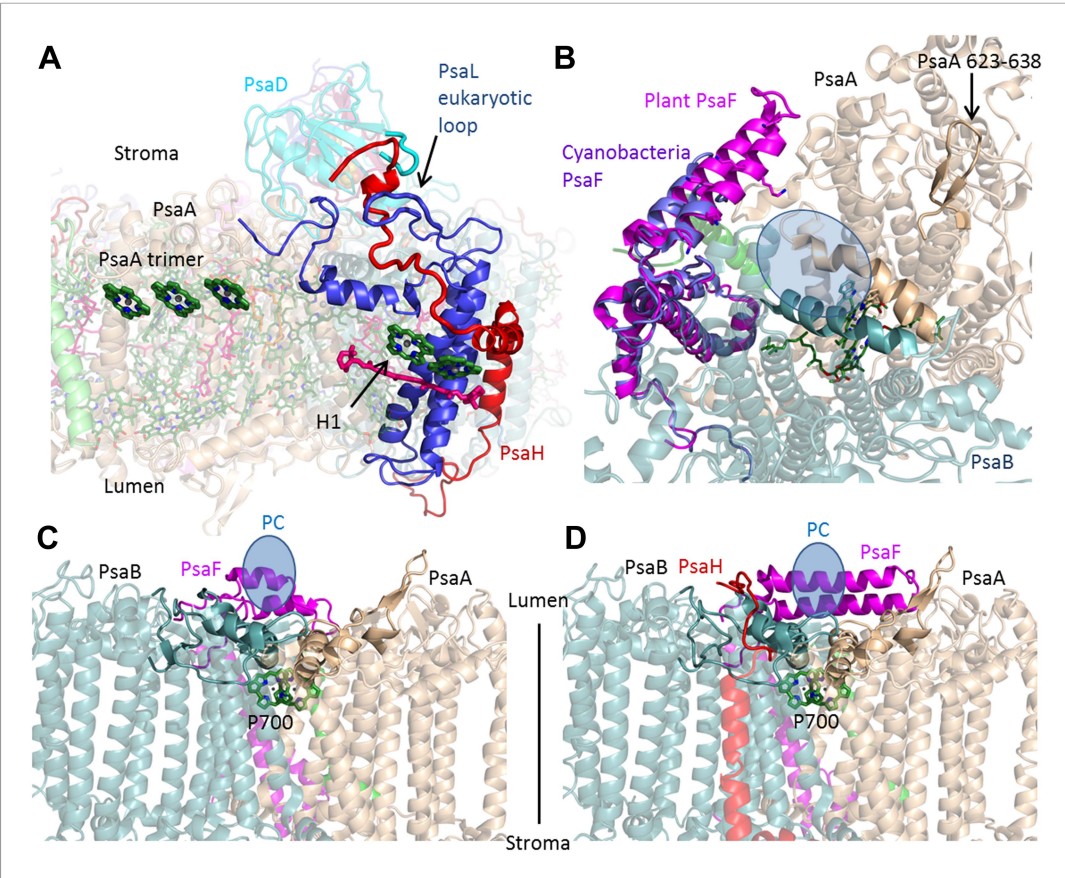

**Figure 3**. PsaH connects the state II and the PC binding sites. (**A**) PsaL is presented in blue, and the path of the PsaH subunit (red) travels through its extended, eukaryotic specific, loop. The new PsaH chlorophyll (green, marked as H1) connects to the PsaL-coordinated chlorophyll and carotenoid (green and pink), as well as an additional chlorophyll trimer (middle left, marked as PsaA trimer), which is proposed to connect PSI to LHCII in state II. (**B**) A luminal view of the PSI surface at the PC binding site. The two hydrophobic helices are colored in grey (PsaA) and light blue (PsaB). The luminal PsaA loop is highlighted on the background of the entire PsaA subunit. All the ligands (with the exception of P700) are not shown. (**C** and **D**) Side by side view of the cyanobacterial (**C**) and plant (**D**) PC binding sites showing the extended PsaF helices that limit access from the membrane plane and the N terminus of PsaH (red), which parallels the configuration of the PsaA luminal loop. The approximate location of PC is indicated in blue.

The following figure supplement is available for figure 3:

**Figure supplement 1**. The proposed configuration of the state II super complex.

Three elements make up the PC binding site: the first is a positive patch located along the helix-turn-helix N-terminal domain of PsaF (*Hippler et al., 1997*, *1998*; *Ben-Shem et al., 2003*). The second element is a hydrophobic patch composed of two parallel helices in PsaA and PsaB (*Figure 2B*) (*Sommer et al., 2006*; *Kuhlgert et al., 2012*). The third conserved feature of the PC binding site is a PsaA luminal loop protruding from the generally flat luminal surface of PSI. This loop is found in both plants and cyanobacteria PSI; the only exception being sequences from marine *Prochlorococcus* and their phages (*Mazor et al., 2012*). As seen in *Figure 3C,D*, the binding site of the cyanobacterial and plant complexes are similar. However, it is clear that the plant binding site is buried deeper in the complex, this is achieved by the extension of the PsaF N-terminal and by the new position of the N-terminus of PsaH, which forms a loop mirroring the conformation of the conserved luminal PsaA loop, suggesting a direct role for PsaH in PC binding.

## Core antenna, red pigments, and excitation energy transfer

The PSI core is a highly efficient hub onto which diverse antennae systems connect such as phycobilisomes and IsiA-like assemblies in cyanobacteria and red algae and LHC type antennae in

eukaryotes (*Berera et al., 2009*; *Engelken et al., 2010*; *Nelson and Junge, 2015*; *Wahadoszamen et al., 2015*). Remarkably, the core pigment organization is conserved across kingdoms despite this diversity in connected antenna (*Amunts et al., 2007*; *Busch and Hippler, 2011*; *Croce and van Amerongen, 2013*), which suggests that the connection points between the core and the antennae are conserved. The existence of red-absorbing pigments (or 'red traps') is a general property of PSI (*Morosinotto et al., 2005*; *Wientjes et al., 2012*). These pigments affect the rate of trapping in PSI and can affect the path of excitation migration in the complex. Most of the eukaryotic red pigments have been shown to reside at LHCI. However, red pigments may be lost from the core complex during the isolation of LHCI. The first high-resolution PSI structure from thermophilic cyanobacteria revealed the organization of the core antenna (*Jordan et al., 2001*). A stacked chlorophyll trimer supported by an extended loop in PsaB was the best candidate for one of the strong red absorbers in this complex (*Jordan et al., 2001*). PsaB sequences from eukaryotes and many cyanobacteria lack this extended loop, resulting in this chlorophyll trimer being lost, as has been shown in the plant and mesophilic PSI structures (*Amunts et al., 2010*; *Mazor et al., 2014*). At the current resolution, we observed new core chlorophyll bound between PsaG and Lhca1 and a newly discovered lipid (*Figure 4A*). This new chlorophyll restores the stacked chlorophyll trimer independent of the shortened PsaB loop and is responsible for one of the connection points between the core complex and the LHCI antenna, with a Mg–Mg distance of 12.5 Å between it and chlorophyll 1010 in Lhca1 (the nomenclature for LHCII is used to describe Lhcas [*Standfuss et al., 2005*]). On the stromal side of the membrane, an additional chlorophyll trimer first discovered in *Synechocystis* is also responsible for an antenna attachment point with a Mg–Mg distance of 13.7 Å between the core chlorophyll A40 and chlorophyll 1005 in Lhca1 (*Figure 4—figure supplement 1*). We suggest that chlorophyll trimers located at the periphery of the core antenna are extremely important for antenna attachment and are probably general attachment points to the core that are utilized not only by eukaryotes, but also by other antenna systems in cyanobacteria.

In contrast to the previous plant structures, which included a small pool of 'Gap chlorophylls', only six pigment pairs connect LHCI to the core antenna in the current structure. Lhca1 is the main connector for excitation transfer, harboring three chlorophylls that are within 14 Å of reaction center pigments (*Figure 4A,B* and *Figure 4—figure supplement 1*). This close proximity ensures efficient and fast energy transfer. Surprisingly, Lhca3 is also one of the main connection points with two such pairs (*Figure 4C*). The final excitonic connection between PSI to LHCI is located between chlorophyll J1302 bound by PsaJ and chlorophyll 2010 (A chlorophyll b molecule). The Mg–Mg distance of this pair is quite large (17.6 Å) however, since the gap between LHCI and PSI allows for some movement, this distance can change to provide an efficient link between the core and LHCI (*Figure 4D*). To summarize, the extremely fast and efficient energy transfer processes that typify PSI-LHCI occurs through only six pairs of pigment molecules located at three sites. These sites connect to the core antennae at the PsaG and PsaK poles through Lhca1 and Lhca3, with Lhca2 playing a relatively minor role.

## The structure of the LHCI complex

Our structure includes the fully modeled LHCI belt with nearly complete structures of all four Lhca proteins. These structures reveal the essential features of the specific interactions between: each Lhca protein and the core; the red chlorophyll assembly present in Lhca4 and Lhca3; and a previously unknown pigment binding site, which is the probable site for the recently discovered non-photochemical quenching (NPQ) at the luminal gap region of LHCI (*de Bianchi et al., 2010*; *Ballottari et al., 2014*).

## Overall view

The LHCI belt is located on the PsaF side of the PSI core. On the stromal face of the membrane, the four conserved N-terminal domains connect each Lhca subunit to its neighbor through interaction with an Lhca-specific loop (loop 23) (*Figure 5A*) that follows immediately after the second transmembrane helix and supports a new chlorophyll site (numbered 16) in Lhca4 and Lhca2 (*Figure 6A,B*). Lhca1, which interacts with PsaG in the core complex, completely lacks this loop and instead contains a short, positively charged linker in this region (*Figure 5A*). One of the major changes in the structure of LHCI is the reversal in the polarity of PsaG. While PsaG occupies roughly an equivalent position compared

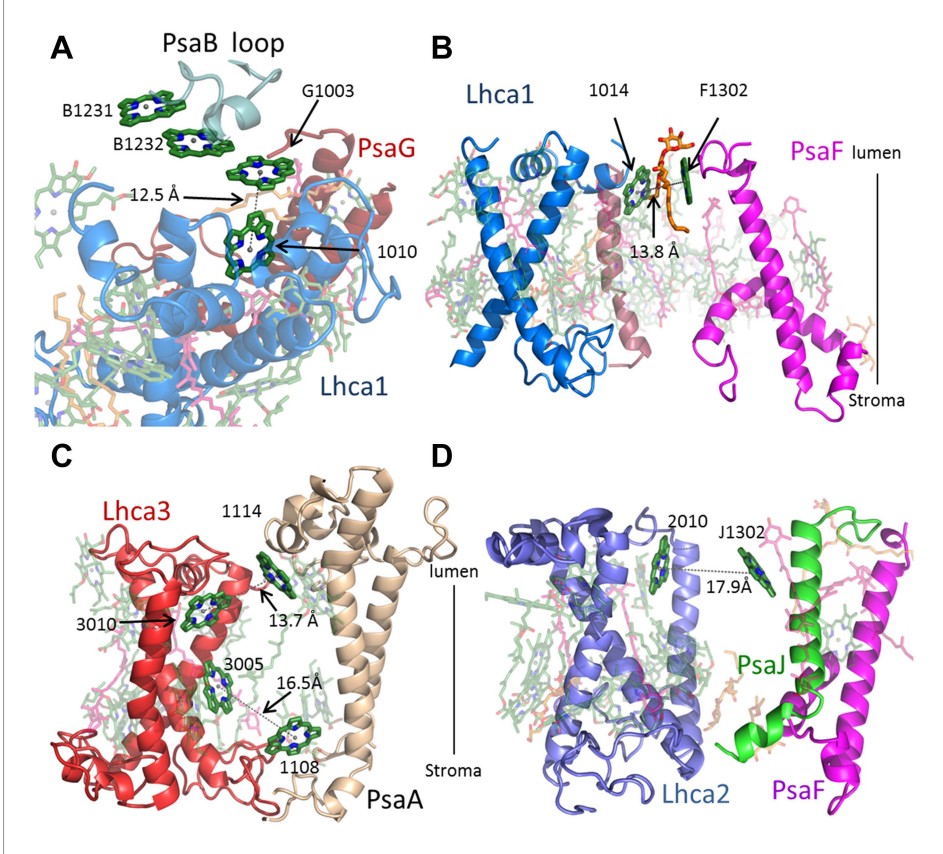

**Figure 4**. Antenna connections in PSI-LHCI. (**A**) The configuration of the PsaG-Lhca1 pigment connection shown from the luminal side of the membrane. The new stacked chlorophyll trimer (numbered B1231, B1232 and G1003) is shown. The N-terminus of PsaG (dark red) supports one of the chlorophylls making up this trimer. The entire trimer is connected with chlorophyll 10 (numbered 1010) in Lhca1 (blue). (**B**) The second LHCI-PSI connection between Lhca1 and PsaF (magenta) on the luminal side of the membrane is bound by a lipid (orange). (**C**) The Lhca3 (red) -PsaA connection. Two chlorophyll pairs mediate this interaction. At the lumen face, 13.7 Å separate chlorophyll 3010 from chlorophyll 1114. On the stromal side, chlorophyll 3005 and chlorophyll 1108 are 16.5 Å apart. (**D**) Lhca2 (blue)—PsaJ (green) connecting chlorophylls.

The following figure supplements are available for figure 4:

**Figure supplement 1**. LhcA1 connects to PSI through chlorophyll trimers.

**Figure supplement 2**. PsaG occupy roughly an equivalent position compared to previous PSI-LHCI structures (PsaG from 2WSC in cyan, PsaG from the current structure in red).

to previous PSI-LHCI structures, the first transmembrane helix contacts Lhca1 while the second one contacts PsaB (*Figure 4—figure supplement 2*). Connections between each Lhca and the core are mediated by small regions immediately preceding the first transmembrane helix and are stabilized by salt bridges between negatively charged residues positioned at the membrane entrance point and a conserved arginine located at a helix turn below them (*Figure 5A*). Additional Lhca–Lhca interactions are provided by a short C-terminal segment on the luminal side that binds helix 2 as it exits the membrane, mainly via hydrophobic interactions (*Figure 5—figure supplement 1*).

Genetic analysis in plants has revealed that each subunit has a specific binding site and, with the exception of the Lhca4-Lhca5 pair, the various Lhcas are not interchangeable (*Lucinski et al., 2006*; *Wientjes et al., 2009*). The specific binding sites are identified in the current structure. At the PsaG pole of LHCI, helix C of Lhca1 interact**s** with the first transmembrane helix of PsaG. Additional protein–protein interactions occurred between a stromal loop of PsaB (aa 307–320) and the N-terminus of Lhca1 (*Figure 5A*).

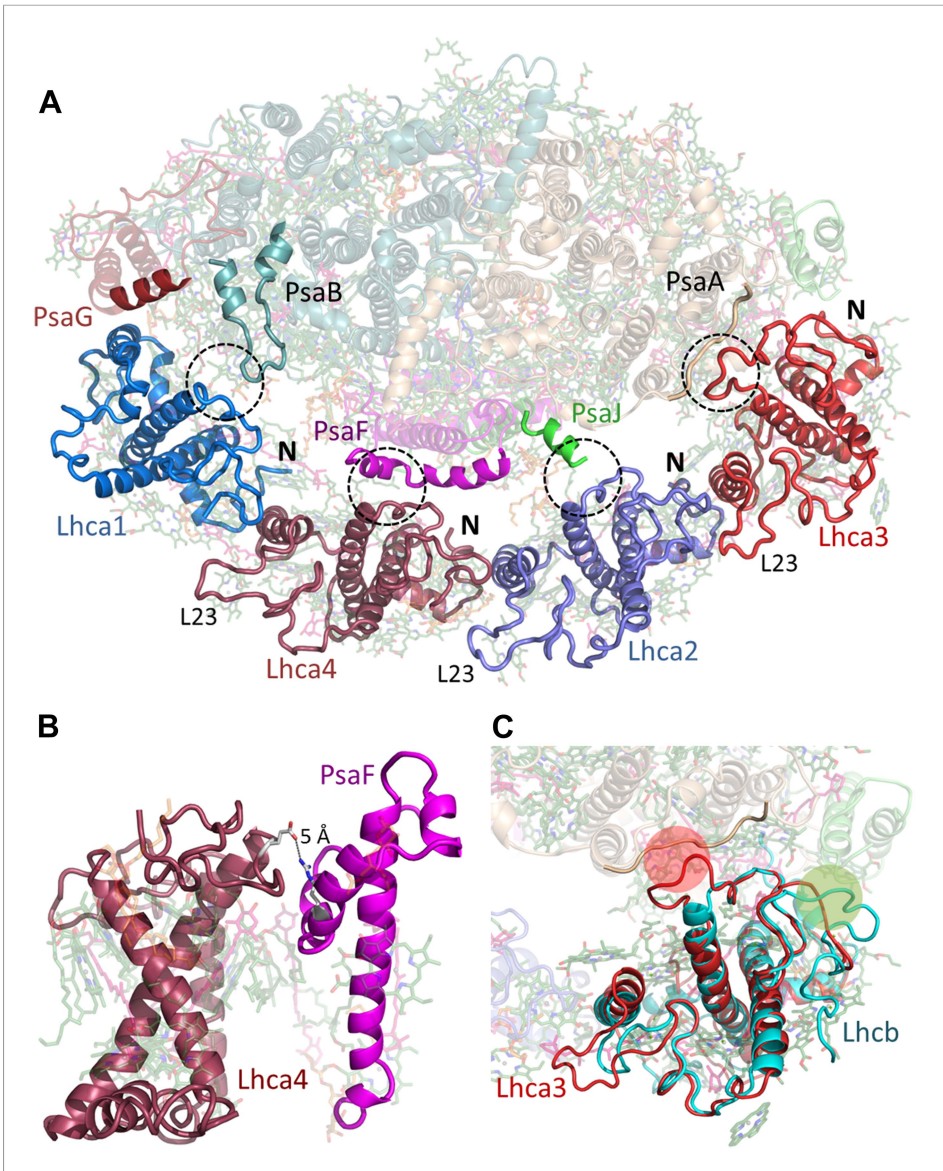

**Figure 5**. Lhcas connect to the PSI core through conserved structural elements. (**A**) A view from the stromal side of the membrane of LHCI. Each Lhca connects to the next one through its conserved N-terminal domain (marked N). All connections to the core are mediated by the short region preceding the entrance of the first transmembrane helix into the membrane (marked by a circle). The extended loop (L23) is conserved in all Lhcas except Lhca1. (**B**) Electrostatic interactions determine the specificity of the Lhca4/5 binding site to PsaF. The conserved E84-R209 interaction occurs within the PsaF (magenta) domain, which is partially membrane-buried, and Lhca4 (raspberry). (**C**) Superposition of Lhca3 (red) and LHCII (cyan). Lhca3 interacts with the core via contacts with PsaA and PsaK through short extension (red circle) or deletion (green circle) in the otherwise highly conserved N terminus. The extension of loop 23 is also seen at the bottom left corner of the image.

The following figure supplement is available for figure 5:

**Figure supplement 1**. Luminal side connections between the core PSI and LHCI.

The next contact point between LHCI and the core occurs between the N-terminus of Lhca4 and the C-terminus of PsaF, the conformation of which is almost identical to the conformation found in cyanobacteria. This interaction consists of hydrophobic patches surrounding charged residues in the membrane-buried regions of both proteins. This binding site should be shared between Lhca4 and

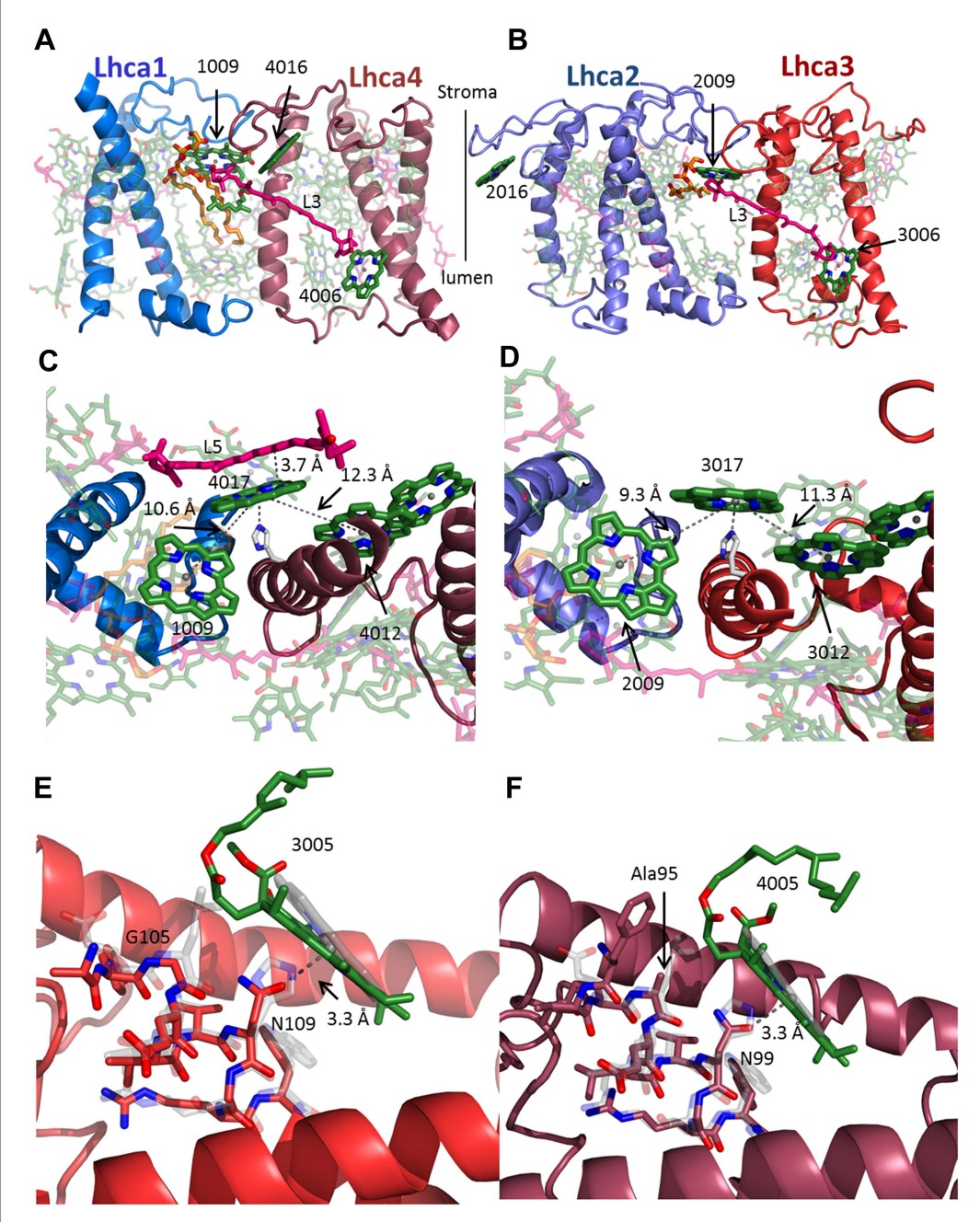

**Figure 6.** The interface within the Lhca heterodimers provides a mechanism for NPQ. (**A** and **B**) Structure of the Lhca1/4 (**A**) and Lhca2/3 (**B**) heterodimer as viewed from the membrane plane extrinsic to the PSI-LHCI complex. The conserved connecting carotenoid in position L3 (pink) serves a structural role, connecting the two subunits of the dimer by linking chlorophyll 6 (bottom right) to chlorophyll 9 (upper left corner). The position of the new chlorophyll site bound by loop 23 of Lhca2 (numbered 2016) is also shown. (**C** and **D**) A new chlorophyll site connects the pigments of the heterodimer. Chlorophyll 17 (numbered 4017 or 3017) is shown as viewed from the luminal side of the membrane. The connections between the red pigment cluster on the right and position 9 in the adjacent subunit are marked with grey lines. The new carotenoid site, L5, is shown in **C**. This site lines the gap between LHCI and PSI and is therefore easily interchangeable. The distance between L5 and of chlorophylls 17 π systems is 3.7 Å. (**E**) Lhca3 is colored in red. A short helix segment from the first transmembrane helix is shown as sticks. Chlorophyll 3005 is part of the red-absorbing dimer. LHCII (PDB: 2BHW) was superimposed on the structure and shown in faint grey. Chlorophyll 612 is shown as a ring. The coordinating side chain was changed from histidine 68 in LHCII to asparagine 109 in Lhca3. Another important change is the leucine to glycine modification at position 105, which

*Figure 6. continued on next page*

*Figure 6. Continued*

allows chlorophyll 3005 to alter its ring orientation by approximately 12°. (**F**) The same site in Lhca4 shown in a similar fashion with Lhca4 colored in raspberry. The same basic coordination is observed, though the ring tilt is smaller because of the alanine occupying position 95.

The following figure supplement is available for figure 6:

**Figure supplement 1**. Identification of chlorophyll b molecules.

Lhca5 and indeed, arginine 209 of PsaF and glutamate 84 of Lhca4 form electrostatic interactions in the middle of the hydrophobic binding site (*Figure 5B*). This residue is conserved in Lhca5 as well, but absent from Lhca1-3 sequences, explaining the specificity of the interaction. The PsaF-Lhca4 interaction in the current structure does not include any pigments. Energy transfer from Lhca4 to the core probably occurs through the Lhca1-PsaB pigment cluster or the Lhca2-PsaJ clusters. Lhca2 is connected to the core almost exclusively through interactions with the N-terminus of PsaJ on the stromal side of the membrane (*Figure 5—figure supplement 1*).

The main contact point of Lhca3 is in the N-terminus of PsaA, which contacts a small patch just before helix 1 enters the membrane as in all other Lhcas (*Figure 5C*). The structure shows that this site is the main determinant of Lhca binding to the core. In contrast to the previous PSI-LHCI model, Lhca3 follows the general fold of LHCII (rmsd 0.8 Å between the two apoproteins). Departures from the LHCII fold are seen in key contact points where small loops were extended or deleted from the otherwise conserved N-terminus domain to facilitate protein–protein interactions with the core (*Figure 5C*).

## Energy transfer and photo-protection mechanisms in Lhca1/4 Lhca2/3 heterodimers

All four Lhcas are remarkably similar to each other and to LHCII. Most of the differences between them can be explained by their interaction partner, such as the loss of loop 23 in Lhca1 due to its binding to PsaG.

Two key differences in pigment organization were found between lhcas and other lhcs. The extended loop 23 supports a new chlorophyll-binding site (numbered 16), common to Lhca4 and Lhca2 (*Figure 6A,B*) and an additional chlorophyll site, coordinated by transmembrane helix two, connects the two partners of each heterodimer.

Chlorophyll b pigments are bound by the different Lhca proteins to various extents and serve as antennae pigments. Four binding sites for chlorophyll b were detected in the high resolution structure of LHCII (*Liu et al., 2004*; *Standfuss et al., 2005*) as well as in the structure of CP29 (*Pan et al., 2011*). We were able to assign nine chlorophyll b sites in LHCI based on electron density maps (*Figure 6—figure supplement 1*). All Lhca proteins contain a glutamine to glutamate change (similarly to CP29) in equivalent positions to position 131 of LHCII, this change makes the binding of Chlorophyll b less likely at three sites (10, 12 and 13). In agreement with this change we find that site 12 is occupied by chlorophyll a in all Lhcas and sites 10 and 13 are occupied differentially. The distribution of chlorophyll b sites in LHCI is markedly uneven, with three sites located in Lhca2 (2010, 2011 and 2013), two sites found at Lhca1 (site 1009 and 1010), three sites at Lhca4 (4010, 4011 and 4013) and a single site on Lhca3 (site 3011). These findings are consistent with mutational data which identified more chlorophyll b sites on Lhca2 then on Lhca3 (*Castelletti et al., 2003*). Site 13 on Lhca1, 2 and 4 is probably a mixed site, which can accommodate both chlorophyll a and chlorophyll b. The high number of chlorophyll b pigments in Lhca2 and the fact that its closest connection to the core is mediated by chlorophyll b (site 2010, shown in *Figure 4D*) is consistent with our suggestion that Lhca1 and Lhca3 are the main junctions for excitation energy transfer from LHCI to the PSI core. The dipole orientations of all but two chlorophylls were identified in the current structure. The most significant changes in LHCI compared to LHCII exist in two sites, 2009 and 4009, where 90° rotations are observed. Such a rotation is expected to impact the energy transfer processes within these LHCI subunits but the significance of this change cannot be ascertained from the rotation alone.

Each of the two Lhca1/4 Lhca2/3 heterodimers contains a blue- and red-absorbing subunit. The structures of Lhca3 and Lhca4 clearly show a coordinating asparagine residue unique to this site (*Arnoux et al., 2009*). The configurations of the pigments themselves differed, with the red

pigments of Lhca3 and Lhca4 tilted approximately 12° relative to their orientation in LHCII, Lhca1, and Lhca2 due to a second change from a bulky leucine residue at position 64 of LHCII to glycine and alanine in Lhca3 and Lhca4, respectively (*Figure 6E,F*). Additional connections that stabilize LHCI are formed by three luteins at position L3. These luteins bridge the chlorophylls at site 6 in Lhca1, Lhca3, and Lhca4 to PsaG, Lhca2, and Lhca1 (*Figure 6A,B*).

Photosynthetic eukaryotes respond to high light conditions by decreasing the efficiency of energy transfer from the antennae in a process called NPQ. First discovered in the PSII complex, NPQ was recently reported in PSI. NPQ sites are located on LHC proteins bound by the hydroxylated carotenoid zeaxanthin, which quenches harmful chlorophyll triplets (*Standfuss et al., 2005*).

The main excitonic connections in the two Lhca dimers appear to be intimately linked to NPQ. A new chlorophyll site (numbered 17) coordinated by a histidine residue (histidine 150 in Lhca4 and histidine 170 in lhca3) is located at the heterodimer interface of both Lhca1/4 and Lhca2/3. Site 17 links the putative red chlorophyll pair with site 9 of the adjacent complex. In the Lhca1/4 interface, a new carotenoid site (numbered 4505 or L5) forms co-planar π systems (plane to plane distance 3.7 Å) with the ring of chlorophyll 4017, positioning it in a configuration that should provide efficient photo-protection from chlorophyll triplets (*Figure 6C,D*). We propose that this is also the site of NPQ in LHCI, and this fits well with the experimental observation (*Standfuss et al., 2005*) that the zeaxanthin responsible for NPQ is located near the red pair in the PSI-LHCI luminal gap region.

In response to lumen acidification, zeaxanthin is synthesized from violaxanthin by violaxanthin deepoxidase (VDE) as part of the xanthophyll cycle. Both the activity and location of VDE are regulated by pH changes (*Ballottari et al., 2014*). At low pH, VDE is activated and binds the luminal side of the membrane, gaining access to its substrate. On the luminal side, a large gap (25 Å) separates LHCI from the core. This gap stems from the fact that most of the connections between LHCI and PSI are located at the stromal side (seen in *Figure 7C*, also compare *Figure 5A* and *Figure 5—figure supplement 1*), leaving the luminal side open.

The complete modeling of LHCI side chains shows that the luminal side of PSI-LHCI contains patches of negative potential distributed on both sides of the PSI-LHCI gap (*Figure 7*). We suggest that the negative charge characteristic of the luminal side of PSI-LHCI is an important feature regulating the accessibility of VDE to the PSI-LHCI gap. Under low light conditions, the relatively high luminal pH will result in strong repulsive forces between these negative patches and the acidic domain of VDE, preventing VDE from binding to PSI. Under high light, acidification of lumen, which is known to trigger VDE activity, will result in partial neutralization of these patches and allow VDE access to the PSI-LHCI gap regions (*Arnoux et al., 2009*) (*Figure 7D*). Carotene L4505 is ideally positioned to be exchanged easily via this mechanism (*Figure 7C*). Ligands, which are easily exchangeable, are particularly labile and may disassociate from the complex during its purification, accounting for a lack of a second carotene bound at the analogous position in Lhca3. In the current structure, position L5 is occupied with zeaxanthin; however, the pigment assignment cannot be definite, even at the current resolution.

In this work we presented the most complete plant PSI-LHCI structure obtained thus far, revealing the locations of and interactions among its protein subunits and more than 200 non-covalently bound photochemical cofactors. Using the new crystal structure, we examined the network of contacts among the protein subunits from a structural perspective, which provide the basis for elucidating the functional organization of the complex.

## Materials and methods

### Purification of plant PSI

Pea seeds (*Pisum sativum* var. Kalvadon) were washed with running water for 6–8 hr and seeded in a vermiculite tray. Plants were germinated in the dark at 22°C for 5 days in shaded sunlight or the dark. After germination, the plants were grown for an additional 10 days under cool-white fluorescent light at a photon flux density of 90–130 µE m$^{-2}$ s$^{-1}$ in a 14 hr light/10 hr dark cycle at 22°C. Approximately 200 g of leaves were ground for 25 s in a blender with 1000 ml of ice-cold solution containing 0.3 M sucrose, 15 mM NaCl, 30 mM Tricine-NaOH (pH 8), 1 mM PMSF, 15 µM leupeptin, and 1 µM pepstatin A. The slurry was filtered through eight layers of cheesecloth and chloroplasts pelleted by centrifugation at 1000 g for 9 min. The pellet was suspended in 500 ml of hypotonic medium (10 mM Tricine-NaOH, pH 8) to disrupt the chloroplasts. Thylakoids were collected by centrifugation at 12,000 g for 10 min and resuspended in 500 ml of buffer containing 10 mM Tricine-NaOH (pH 8) and

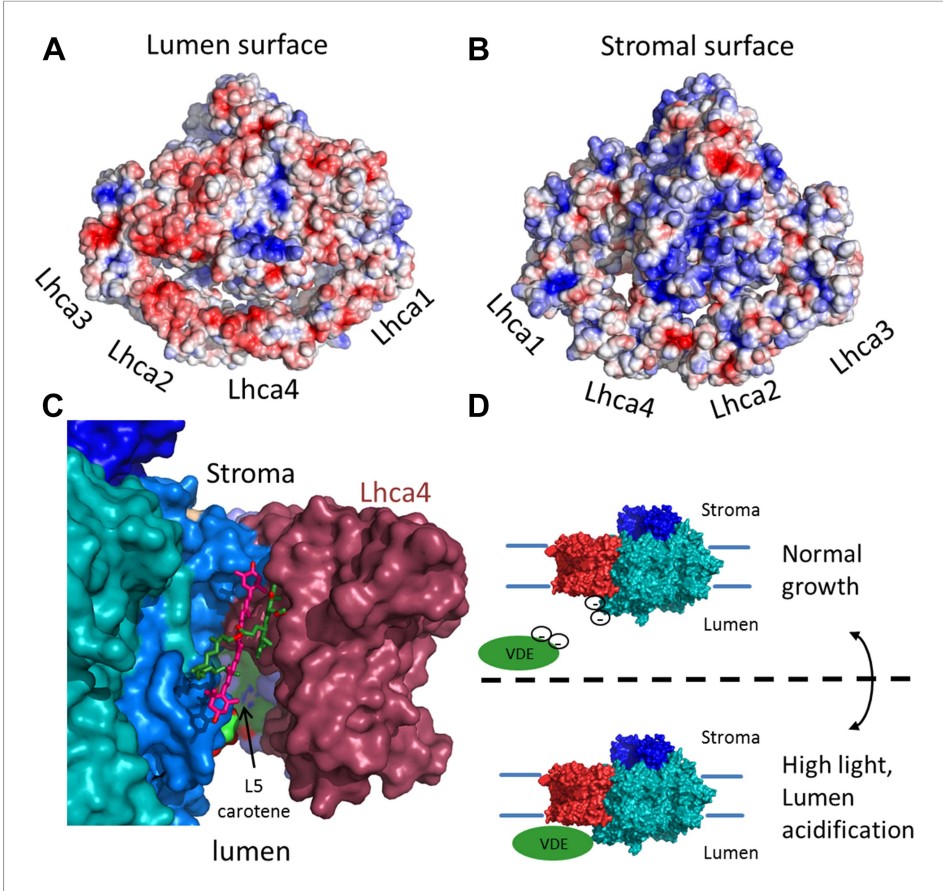

**Figure 7**. The surface electrostatic potential of PSI-LHCI. (**A**) Luminal view of the electrostatic potential (red for negative and blue for positive charges) generated from PSI-LHCI apoproteins. The gap region is shown as an open cavity (the ligands occupying this cavity were omitted from the calculation) lined with negative patches. The plastocyanin binding site can be distinguished as a blue patch generated by the positively charged luminal double helix domain of PsaF. (**B**) The stromal electrostatic surface of PSI-LHCI showing the putative ferredoxin binding site as a basic patch on the left side of the stromal ridge. (**C**) A side view of the stromal gap shows the large opening facing the lumen. Lhca1 and PsaG are omitted to reveal the internal cavity. This surface was drown around all ligands identified in the gap with only chlorophyll 4017 and carotene 4505 omitted. (**D**) We propose that the negative patches lining the gap play a role in preventing the accessibility of VDE to gap pigments during regular growth. While at high light conditions, lumen acidification partially neutralizes these negative charges, allowing VDE activity on gap xanthophylls.

150 mM NaCl. The thylakoid membranes were then pelleted at 8,000 g for 10 min and resuspended in a minimal volume of STN2 buffer (0.4 M sucrose, 20 mM Tricine-NaOH, pH 8). The thylakoid concentration was adjusted to 3 mg of chlorophyll per ml and 0.4% n-dodecyl-α-D-maltoside (DDM) added. This concentration of detergent selectively extracts the ATP synthase, b6f, and PSII complexes. After 5 min incubation on ice, the detergent- treated thylakoid membranes were collected by ultra-centrifugation at 200,000 g for 30 min. The pellet was resuspended in a minimal volume of STN2 buffer, adjusted to 3 mg chlorophyll per ml, and stored at −80˚C.

Frozen thylakoid membranes (20–25 ml) containing 3.0 mg of chl/ml were thawed in cold water and solubilized with 1.5% DDM. Insolubilized material was removed by ultracentrifugation at 120,000 g for 15 min. The supernatant was applied to a DEAE-cellulose column (DE-52, Whatman, Inc., 1.5 × 18 cm) pre-equilibrated with 15 mM Tricine-Tris (pH 8.0) containing 0.25% DDM. The column was washed with the same buffer and PSI eluted with a 0–230 mM tetraethylammonium chloride linear gradient (75 ml in each chamber) in 15 mM Tricine-Tris (pH 8.0) containing 0.25% DDM. Dark green fractions containing PSI were precipitated by 10% PEG6000 (Hampton Research, Aliso Viejo, CA), followed by centrifugation at 5,000 g for 6 min. The pellet was dissolved in 15 mM Tricine-Tris (pH 8.0) and 0.05%

**Table 1**. Data collection and refinement statistics

**Data collection**

| | | | |
|---|---|---|---|
| Beamline | BESSY PX14.1 | SLS PXI–X06SA | ESRF ID 23-2 |
| Wavelength (Å) | 1.73 | 1.74 | 0.873 |
| Resolution (Å) | 40–2.8 (2.85–2.8) | 40–2.8 (2.85–2.8) | 50–3 (3.1–3) |
| Space group | $P2_12_12_1$ | $P2_12_12_1$ | $P2_1$ |
| **Unit cell dimensions** | | | |
| a, b, c (Å) | 188.7, 200.8, 212.4 | 188.6, 201.3, 212.7 | 120.6, 189.2, 129.7 |
| α, β, γ | 90, 90, 90 | 90, 90, 90 | 90, 91.1, 90 |
| Measured reflections | 7,831,302 | 14,841,310 | 724,010 |
| Unique reflections | 198,911 | 200,218 | 116,039 |
| Rpim (%) | 0.051 (1.276) | 0.030 (0.180) | 0.099 (0.812) |
| $<I/\sigma(I)>$ | 10.5 (1.2) | 13.4 (1.4) | 5.8 (1.3) |
| Completeness (%) | 99.9 (98.4) | 99.9 (98.4) | 99.9 (99.2) |
| Redundancy | 39.9 (37) | 74.1 (33.4) | 6.2 (5.4) |
| **Refinement statistics** | | | |
| Resolution (Å) | 40–2.8 | 40–2.8 | 50–3 |
| Rwork/Rfree | 25.6/26.5 | 24/25.2 | 25.8/29.3 |
| No. of chains | 16 | 16 | 17 |
| No. of ligands | 214 | 214 | 197 |
| Average B-factor (Å²) | 98.7 | 112 | 96.6 |
| **R.M.S deviations** | | | |
| Bond angles | 1.9 | 2 | 2.4 |
| Bond lengths | 0.004 | 0.005 | 0.011 |
| **Ramachandran statistics** | | | |
| Favoured region % | 90.2 | 90.2 | 86.6 |
| Allowed region % | 7.1 | 7.1 | 8.7 |
| Outlier region % | 2.7 | 2.7 | 4.7 |

Data collection, scaling and merging statistics were calculated using XDS, AIMLESS and PHENIX XTRIAGE. Refinement statistics are from PHENIX.

DDM. The green solution was applied to a 10–35% sucrose gradient containing the same buffer and centrifuged using the SW-40 rotor (Beckman Coulter, Fullerton, CA) at 37,000 rpm (170,000 g) for 16 hr. The wide green band containing PSI was collected and loaded onto a second DEAE-cellulose column (0.5 × 4 cm) pre-equilibrated with 15 mM Tricine-Tris (pH 8.0) and 0.05% DDM, mainly to concentrate it for a second sucrose gradient. PSI was eluted with 230 mM tetraethylammonium chloride. The collected dark green fraction was applied to a 10–35% sucrose gradient and centrifuged at 57,000 rpm (330,000 g) for 4 hr using an SW-60 rotor (Beckman Coulter). Purified PSI appeared as a dark band in the middle of the tube, but only the central part of the band was collected. The material was precipitated with 15% PEG1500 and 100 mM tetraethylammonium chloride and centrifuged at 10,000 g for 4 min. The pellet was dissolved in a solution containing 2 mM Tricine (pH 8.75) and 0.02% n-dodecyl-β-D-thiomaltoside (DTM, Glycon Biochemicals, Luckenwalde, Germany) and adjusted to a chlorophyll concentration of 2.5 mg/ml.

## PSI crystallization and cryogenic protection
Crystallization was performed manually in 24-well plates using the sitting drop variant of the vapor-diffusion technique at 4°C (Charles Super Company, Natick, MA). Aliquots (6–8 μl) of PSI solution were mixed with equal volumes of reservoir solution (50 mM di-potassium phosphate, 50

**Table 2**. Amino acid changes between PSI-LHCI structures

| Subunit | Number of amino acids | Modeled amino acids | | Number of changes |
|---------|----------------------|------|------|------------------|
| | | 4Y28 | 2WSC | |
| PsaA | 758 | 741 | 729 | 1 |
| PsaB | 734 | 732 | 732 | 1 |
| PsaC | 81 | 80 | 81 | 1 |
| PsaD* | 156 | 140 | 135 | 5 |
| PsaE* | 92 | 68 | 65 | 9 |
| PsaF* | 154 | 150 | 154 | 18 |
| PsaG* | 98 | 95 | 95 | 15 |
| PsaH* | 95 | 84 | 69 | 11 |
| PsaJ* | 42 | 41 | 42 | 5 |
| PsaK* | 134 | 79 | 85 | 10 |
| PsaL* | 168 | 160 | 161 | 21 |
| Lhca1* | 204 | 193 | 165 | 12 |
| Lhca2 | 256 | 206 | 176 | 3 |
| Lhca3 | 242 | 210 | 162 | 16 |
| Lhca4 | 252 | 197 | 166 | 3 |

The number of modeled amino acids in each subunit is shown and compared to the most recent PSI-LHCI structure (2WSC). Insertions, deletions and extensions are counted as a single change. Since the genome sequence of *Pisum Sativum* is not completely known (genes with no DNA data are marked with an *) we relied on high-throughput mRNA sequence data for verification (**Franssen et al., 2011**).

mM Tris [pH 8], 12–17% PEG400, 1% glycerol, 2 mM L-glutathione, and 0.03% octyl glucose neopentyl glycol) and equilibrated against 0.5 ml of reservoir solution. Dark green rectangular crystals appeared after 3 days at the higher PEG concentrations, but the best diffracting crystals appeared after 1 month at the lower PEG concentrations.

For cryogenic protection, the crystals were moved to a solution containing 50 mM di-potassium phosphate, 50 mM Tris (pH 8), 20% PEG400, 2% glycerol, and 2 mM L-glutathione. After a brief incubation the crystals were soaked sequentially in the same buffer containing 5% and 10% glycerol and immediately frozen in liquid nitrogen. X-ray diffraction data were collected at the European Synchrotron Radiation Facility (ID23- 2, ID23-1, ID-29 and MASSIF-3), the Swiss Light Source (PXI, PXII, and PXIII), and BESSYII.

## Data collection and processing

Images were collected at 0.1° oscillation using full beam at exposures of 0.1–0.05 s. The large size of the crystals helped mitigate the effect of radiation damage, yet only 60°–90° of data were collected from each crystal using constant translation of the crystal in the beam. Images were integrated using XDS (**Kabsch, 2010**) and scaled with XSCALE or AIMLESS (**Evans and Murshudov, 2013**). Measurements were carried out at the peak of the iron fluorescence scan. Individual datasets generally had I/SIGMA values of ~1 at 3.1 Å with CC1/2 values of ~0.5 calculated by XDS. To obtain a more accurate measure of weak reflections, we combined several datasets measured under similar conditions. This combination was possible because of the consistency of the new crystals. The most effective method for combining datasets was to simply choose the sets with the strongest statistics at lower resolutions, and these crystals were also very similar to each other in terms of their unit cell dimensions. The unified datasets contained 40–100 independent measurements of each reflection with I/SIGMA of ~1.5 at 2.8 Å and CC1/2 of ~0.4 at the same resolution (**Table 1**). The CCanomalous calculated by XDS or AIMLESS was ~0.3 at 6 Å. Some individual datasets had measurable anomalous signals (CCanomalous > 0.3) to 4.5 Å, but the final quality of phases was similar, as

judged by the figure of merit for substructures from PHASER runs (typically ~0.6 [*McCoy et al., 2007*]) and visual inspection of the maps.

## Phasing and refinement

The crystals were initially solved by molecular replacement (MR) with a partial model containing only the reaction center subunits with chlorophylls modeled as rings using PHASER. This solution was used to place the three iron-sulfur clusters, which were subsequently used as the initial substructure for locating additional sites. These initial runs located approximately 40 sites of the 100 that were eventually modeled. Phases were improved using DM (*Cowtan, 1994*), and the resulting maps showed most of the transmembrane helices of the reaction center with 11 transmembrane helices of LHCI (missing helix 2 of Lhca3). To improve the phases, information from the visible parts of the reaction center was incorporated as a partial MR solution. This model included the 22 transmembrane helices of PsaA and PsaB, the PsaC subunit, and chlorophyll rings, omitting all of the loops from the proteins. From the maps generated in this step, we proceeded to build the model using Coot (*Emsley et al., 2010*) and used the modified phases as restraints during refinement in either PHENIX (*Adams et al., 2010*) or REFMAC (*Murshudov et al., 1997*). At various points during the refinement process, the phases were recalculated with the newly modeled sites using PHASER and including the improved model (*Read and McCoy, 2011*). Final runs identified 99 sites, 89 of them present in the model. The final model refined to an R-free of 25.2% with 3% Ramachandran outliers, a considerable decrease from previous values (*Table 1*). A complete model for the lhca subunits of LHCI was obtained, as well as extensions and modifications to some of the other PSI subunits (*Table 2*). Images were created using Pymol and electrostatic surfaces calculated using APBS (*Baker et al., 2001*).

## Acknowledgements

The authors would like to thank the ESRF, SLS and BESSYII synchrotrons for beam time and the staff scientists for excellent guide and assistance as well as to the MASSIF beamline team in ESRF. We would like to thank Prof Martin Kupiec and Dr Ofer Rog for their critical reading of the manuscript. This work is supported by a grant no. 293579—HOPSEP from the European Research Council, The Israel Science Foundation through grant No. 71/14 and by the I-CORE Program of the Planning and Budgeting Committee and The Israel Science Foundation (grant No 1775/12).

## Additional information

### Funding

| Funder | Author |
| --- | --- |
| European Research Council (ERC) | Yuval Mazor, Nathan Nelson |
| Israel Science Foundation (ISF) | Anna Borovikova |

The funders had no role in study design, data collection and interpretation, or the decision to submit the work for publication.

### Author contributions

YM, NN, Conception and design, Acquisition of data, Analysis and interpretation of data, Drafting or revising the article; AB, Acquisition of data, Drafting or revising the article

## Additional files

### Major dataset

The following dataset was generated:

| Author(s) | Year | Dataset title | Dataset ID and/or URL | Database, license, and accessibility information |
| --- | --- | --- | --- | --- |
| Mazor Y, Brovikov A, Nelson N | 2015 | TBD | http://www.rcsb.org/pdb/search/structidSearch.do?structureId=4Y28 | Publicly available at RCSB Protein Data Bank (Accession No: 4Y28). |

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
