## [Decision Letter]

Thank you for submitting your work entitled “The structure of plant photosystem I super-complex at 2.8 Å resolution” for peer review at *eLife*. Your submission has been favorably evaluated by John Kuriyan (Senior editor) and three reviewers, one of whom is a member of our Board of Reviewing Editors.

The reviewers have discussed the reviews with one another and the Reviewing editor has drafted this decision to help you prepare a revised submission.

The following individuals responsible for the peer review of your submission have agreed to reveal their identity: Werner Kühlbrandt (Reviewing editor); Petra Fromme (peer reviewer). A further reviewer remains anonymous.

Summary:

In their 2.8 Å structure of pea PSI structure, the authors provide numerous new details of this massive membrane protein supercomplex. In a crystallographic tour de force, the diffraction data, obtained with a new, more highly symmetrical and better ordered crystal form, were phased by the weak Fe, S and P anomalous signal, resulting in a considerably more accurate structure. Overall, the structure is now largely identical to cyanobacterial PSI, although there are significant and important differences, in particular with respect to the functional connections to the 4 Lhca subunits, which are confined to plants.

New features of the 2.8 Å structure include more than 30 previously unidentified chlorophylls, a doubling in the number of resolved carotenoids, which are functionally important, and 4 lipids.

A chlorophyll that presumably plays a role in energy transfer from the antenna to the PSI core was newly identified in the plant-specific subunit H, and details of the interaction site with the electron carrier plastocyanin are resolved, as are the contact points with all 4 LHC1 complexes, with details of the pigments involved.

Essential revisions:

1) It would be important to know how much of polypeptide sequence was actually traced compared to the previous 3.4 Å structure. How many side chain positions needed to be corrected?

2) An understanding of the energy transfer requires a consideration and discussion of the orientation of the dipole transition moments. If these are included, the manuscript gains considerably. How many chl orientations were unambiguously defined at 2.8 Å compared to the previous 3.4 Å structure?

3) How sure is the assignment of chlorophylls to chl a or b? Electron densities and omit maps of typical chls a and b should be shown as a supplementary figure.

4) The proposed mechanism for non-photochemical quenching in PSI is difficult to follow and should be spelt out more simply and clearly, and a schematic drawing to explain it should be added. What is the experimental evidence that the proposed mechanism is correct?

Figure 3: explain in more detail how the model differs from the previous PSI structures.

In the subsection headed “Core antenna, red pigments and excitation energy transfer” and in the fourth paragraph of the subsection headed “Overall view”: is this really a “domain”? It looks like an N-terminal segment.

---

## [Author Response]

*1) It would be important to know how much of polypeptide sequence was actually traced compared to the previous 3.4 Å structure. How many side chain positions needed to be corrected*?

We added a table (Table 2) describing the number of changes (compared to the previously published PSI-LHCI) made in each subunit, as well as the number of modeled amino acids in the final structure. The complexity of the structure and the fact that the highly complex genome of *Pisum Sativum* is not available makes it difficult to ascertain the sequences of each subunit. However the available transcriptome covers all the sequences of PSI subunits. In case of more than one gene encoding single subunit we chose them according to the electron density.

*2) An understanding of the energy transfer requires a consideration and discussion of the orientation of the dipole transition moments. If these are included, the manuscript gains considerably. How many chl orientations were unambiguously defined at 2.8 Å compared to the previous 3.4 Å structure*?

We have determined the orientation of all but two chlorophylls in the final model. Among them 5 chlorophylls in the core complex and 15 in the LHCI changed their orientation, compared to the previous PSI-LHCI structure. These facts are incorporated in the text.

*3) How sure is the assignment of chlorophylls to chl a or b? Electron densities and omit maps of typical chls a and b should be shown as a supplementary figure*.

We have included an example of the electron density maps for chlorophyll b (Figure 6—figure supplement 1), including a simulated annealing omit maps that clearly show the presence formyl-7 group of chlorophyll b.

*4) The proposed mechanism for non-photochemical quenching in PSI is difficult to follow and should be spelt out more simply and clearly, and a schematic drawing to explain it should be added. What is the experimental evidence that the proposed mechanism is correct*?

The experimental evidences are given in references 25 and 44. We do not argue for or against NPQ in PSI. We merely point to the existence of a possible unique site at the luminal gap region of LHCI. In our view this unique site is interesting from the structural and probably functional aspects of LHCI subcomplex. We expanded Figure 7 to better visualize the gap and the new pigment site, as well as a simple model to graphically illustrate our intent.

.

Figure 3*: explain in more detail how the model differs from the previous PSI structures*.

The improved resolution led to a reversal in polarity of the PsaG subunit and we have added a figure to show this (Figure 4—figure supplement 2). The position of PsaK was changed, probably due to different crystal contacts, resembling the cyanobacterial one. It points to the possibility that a Lhcb monomer can bind to PsaA in a similar fashion that Lhca1 binds PsaB. It demonstrates that PsaH binds to four different subunits that may be important attribute for the known stability of plant PSI. The position and orientation of many pigment molecules are now well defined.

*In the subsection headed “Core antenna, red pigments and excitation energy transfer” and in the fourth paragraph of the subsection headed “Overall view*”*: is this really a* “*domain*”*? It looks like an N-terminal segment*.

We refer to three domains in the text: the helix-turn-helix N-terminal domain of PsaF, the N-terminal domain of Lhca/LHCII and the negatively charged domain of VDE. The N terminal domain of Lhca/LHCII is a highly conserved 30 amino acid stretch that folds similarly in the case of LHCII and in all four Lhcas and appears to be important for subunit interaction in all of these cases.